# Using lagged dependence to identify (de)coupled surface and subsurface soil moisture values

Coleen D.U. Carranza[1], Martine J. van der Ploeg[1], and Paul J.J.F. Torfs[2]

[1]Soil Physics and Land Management Group, Wageningen University, the Netherlands
[2]Hydrology and Quantitative Water Management Group, Wageningen University, the Netherlands

*Correspondence to:* Coleen Carranza (coleen.carranza@wur.nl)

**Abstract.** Recent advances in radar remote sensing popularized the mapping of surface soil moisture at different spatial scales. Surface soil moisture measurements are used in combination with hydrological models to determine subsurface soil moisture values. However, variability of soil moisture across the soil column is important for estimating depth-integrated values as decoupling between surface and subsurface can occur. In this study, we employ new methods to investigate the occurrence of (de)coupling between surface and subsurface soil moisture. Using time series datasets, lagged dependence was incorporated in assessing (de)coupling with the idea that surface soil moisture conditions will be reflected at the subsurface after a certain delay. The main approach involves the application of a distributed lag non-linear model (DLNM) to simultaneously represent both the functional relation and the lag structure in the time series. The results of an exploratory analysis using residuals from a fitted loess function serve as *a posteriori* information to determine (de)coupled values. Both methods allow for a range of (de)coupled soil moisture values to be quantified. Results provide new insights on the decoupled range as its occurrence among the sites investigated is not limited to dry conditions.

## 1 Introduction

Although recent decades have seen great advances in remote sensing applications for mapping surface soil moisture (Jackson, 1993; Njoku et al., 2003; Mohanty et al., 2017), most hydrological studies that make use of soil moisture data require integrated values over a certain soil depth (Brocca et al., 2017). Extrapolation of surface soil moisture from remote sensing techniques to depths beyond the sensor's capacity (up to 5 cm) is not a trivial task given the spatio-temporal variability of soil moisture. The vertical distribution of soil moisture, which determines integrated soil moisture content over a soil column, is rarely uniform as more pronounced dynamics are expected closer to the surface compared to deeper in the soil (Hupet and Vanclooster, 2002). Currently, information derived from remote sensing are assimilated into hydrological models to obtain integrated soil moisture values (Houser et al., 1998; Das et al., 2008). However, Kumar et al. (2009) stressed that it is important to assess vertical variability, especially the strength of coupling between surface and subsurface soil moisture for improvement of data assimilation results. Analyses of vertical soil moisture distributions also have important implications for modeling studies, as they could be used for calibration or validation of model parameters (De Lannoy et al., 2006).

The amount of soil moisture at any given time is controlled by factors operating a different time scales. While prevailing atmospheric conditions directly affect surface layers and control the temporal dynamics of soil moisture (Albertson and Montaldo, 2003; Koster et al., 2004), it is the downward movement of water from the surface that dictates the amount of subsurface soil moisture at a given time (Belmans et al., 1983; Rodriguez-Iturbe et al., 1999). Flow rates to the subsurface are driven by hydraulic properties, which are in turn controlled by physical soil characteristics such as texture, bulk density, and structure. Relative to changes in atmospheric conditions, soil physical properties change over longer timescales. Vegetation further modifies vertical soil moisture distribution by root water uptake (Yu et al., 2007) and by changing soil structure (Angers and Caron, 1998).

Given the variability along a soil column, during which conditions do surface soil moisture reflect subsurface soil moisture? Several studies have investigated this relation to address the correspondence between surface and subsurface soil moisture. One of the earliest studies is by Capehart and Carlson (1997) wherein they compared modeling outputs with remote sensing measurements. Using very shallow depths of 5 mm and 5 cm, they referred to decoupling as the deviation from a linear correlation between these depths due to variable drying rates. Further assessment of decoupling from model-generated time series soil moisture data have been investigated using cross-correlation values (Martinez et al., 2008; Mahmood et al., 2012; Ford et al., 2014). High correlation to the subsurface was obtained using lagged values of surface soil moisture. However, cross-correlation is limited to providing a single value throughout the range of soil moisture encountered per lag. Furthermore, cross-correlation generally aims to evaluate the strength of lagged linear dependence between two variables (Shumway and Stoffer, 2010). However, lagged dependence between surface and subsurface soil moisture may not be linear given that non-linear processes determine water flow along the soil profile. Using in situ field measurements, Wilson et al. (2003) investigated spatial surface (0-6 cm) and subsurface (0-30 cm) soil moisture distribution by calculating statistical metrics and by means of a variogram. Decoupling between the two depths was observed which they suggested to be influenced by vegetation, especially root density at surface soil. Their results were also affected by the dry soil moisture range and emphasized the importance of distinguishing between surface and total soil moisture for future applications of remote sensing to atmospheric studies.

Based on previous studies, the term decoupling refers to a weak dependence between soil moisture contents at the surface and subsurface. Recognition of decoupling is important, however most studies have been limited to providing qualitative characterization of conditions when decoupling occurs (e.g. dry period). Only Capehart and Carlson (1997) identified a mid-range soil moisture ($\sim$0.3 $cm^3cm^{-3}$) when the surface and very near surface begin to decouple. Their results, however, are limited to a thin layer of the soil column. In this paper, our main objective is to quantitatively identify a range of surface soil moisture values that is decoupled from the subsurface. Furthermore, we consider depths greater than those investigated by Capehart and Carlson (1997). The ability to quantify (de)coupled surface and subsurface soil moisture contents will contribute to more effective estimation of depth-integrated soil moisture data using remote sensing methods and improved data assimilation results in hydrological models.

We utilized in situ time series datasets at depths of 5 cm and 40 cm to represent surface and subsurface, respectively. Values outside the decoupled range are considered coupled since soil moisture is inherently bounded up a maximum value equal to soil's porosity. The investigation of (de)coupling is based on the idea that surface conditions will be reflected at the

subsurface after a certain delay indicating strong coupling between the two zones, and vice versa. More focus is given to the decoupled soil moisture range since it has greater implications for extrapolation of surface soil moisture values to deeper soil layers. We applied statistical methods to identify conditions of decoupling with no prior assumptions on the type of functional relation between surface and subsurface. As an exploratory step, we first assessed the dependence without considering lags using regression and residuals analysis. The main approach for assessing decoupling was application of distributed lag non-linear models (Gasparrini et al., 2010) to incorporate both the lag structure and the functional relation between surface and subsurface soil moisture. Applications of distributed lag models to econometrics and environmental epidemiology have been well documented (Almon, 1965; Zanobetti et al., 2002; Bhaskaran et al., 2013; Wu et al., 2013). However, their application to hydrological studies have rarely been explored.

INSERT Fig.1 here

## 2   Description of datasets and study sites

Four time series datasets from the Twente soil moisture and temperature monitoring network (Dente et al., 2011) were used in this study (fig.1). Datasets from 2014-2016 are available with only short periods of missing data. The stations are located in agricultural fields with sensors installed at 5 cm, 10 cm, 20 cm, and 40 cm depths. To investigate decoupling, only the 5 cm and 40 cm depths were considered because the largest possible distance was desired. Each station consists of EC-TM ECH2O capacitance probes (Decagon Devices, Inc., USA) that logged soil moisture data every 15 minutes. A calibration procedure using gravimetric measurements was applied prior to analysis (Dente et al., 2011).

Land cover in the area varies from corn in one field (SM05), to grass in two fields (SM05 and SM13), to a forest area (SM20). Values at 40 cm capture the root zone of vegetation for each site. In reality, rooting depths vary and depend on species composition, climate, and plant growth rate. However, the depth considered would still allow for approximation of root zone conditions. The landscape is characterized by flat to slightly sloping terrain. It is important to note that SM20 is located at the eastern foot of a small hilly terrain. Throughout the study period, either land cover remained unchanged or the same crop was planted. The soil types for the stations range from coarse sandy soils to weakly silty soils (Wosten et al., 2013). A summary of the land cover and relevant characteristics of the stations are summarized in Table 1.

Soil moisture values were averaged into daily values to match the available daily rainfall data from the Dutch national weather service (KNMI). For SM13 and SM20, there are some missing data from the beginning of 2014. The datasets from SM13 begins on April 25, 2014 while SM20 begins on May 2, 2015 (fig.2).

INSERT Fig.2 here

INSERT Table 1 here

## 3 Methods

### 3.1 Regression and residuals analysis

As an exploratory step, the dependence between surface and subsurface soil moisture was initially visualized using scatterplots. Conditional means for every 0.01 cm$^3$cm$^{-3}$ interval and vertical bars representing $\pm$ standard deviation were added to show changes in vertical variability across the soil moisture range. Longer standard deviation bars indicate higher vertical variability (fig.3, left). We referred to vertical variability as the uneveness or irregularity in the soil moisture distribution within a certain depth of interest along the vertical profile, in this case up to depths of 40 cm. For the rest of this paper, variability will refer to vertical soil moisture variability, unless otherwise stated. The plotted points were colored per month to show any impacts of seasonality. The effect of rainfall was included by adjusting the sizes of the points proportional to rainfall intensity measured from the nearest KNMI stations. For the overall measure of dependence, Spearman's rank correlation coefficient $R_s$ was computed for every pair of ranked values in the time series. This was chosen as the assumption of linear dependence was not made.

A flexible non-parametric locally weighted regression function (commonly called a loess function, Cleveland and Devlin (1988)) was fitted along the soil moisture range. This was used to explore and identify trends across the soil moisture range. A linear regression was also fitted only for comparison. Residuals were analyzed further for variability not captured by the fitted function (fig.3, right). The residuals variance for every 0.1 cm$^3$cm$^{-3}$ interval as well as the resulting cumulative residuals variance were analyzed to examine variability across the range. The degree of variability was related to the slope of the cumulative variance line, with steep slopes indicating high variability. In addition, a significant change in variance between two points was indicated by a significant change in the slope of the line. The soil moisture value where a significant change in slope occurred was marked by $\theta_c$, this divides the soil moisture range into two groups. The group with a steeper slope was interpreted as the decoupled range, and vice versa. Since the measured variance is sensitive to sample size, a correlation coefficient was calculated to determine if there was significant dependence between the two variables. Residuals variance were first normalized from 0-1 because of the varying soil moisture range encountered at each station.

Results of the exploratory methods were considered *a posteriori* knowledge for analysis of lagged dependence and interpretation of results.

INSERT Fig.3 here

### 3.2 Analysis of Lagged Dependence

#### 3.2.1 Cross correlation

Since decoupling is based on the strength of lagged dependence, the existence of lag between surface and subsurface soil moisture values was first determined. Cross-correlation is known to be a quick and easy method to apply for this objective. Lagged values of surface soil moisture were correlated with instantaneous values at the subsurface. A maximum cross-correlation at negative lags indicated that surface soil moisture is leading subsurface soil moisture, and vice versa (Shumway and Stoffer,

2010). A 10-day lag was deemed long enough to show the presence of lag-lead relations in the time series since the maximum correlation occurred within this period.

### 3.2.2 Distributed lag non-linear model

We incorporated delayed or lagged effects in evaluating the relation between surface and subsurface values, and eventually in determining the (de)coupled values. It should be emphasized that the analysis was primarily focused on examining the trends and relation between surface and subsurface soil moisture. Moreover, it was not intended to replace other existing models for estimating soil moisture or examining its patterns.

A distributed lag non-linear model (DLNM) developed by Gasparrini et al. (2010) was applied to the 5 cm and 40 cm time series datasets at the study sites. Briefly, the model is capable of simultaneously representing both functional the dependence and delayed response between exposure and response values. We considered surface soil moisture as the exposure values that produced delayed effects to the response values at the subsurface. A non-linear model was selected in order to capture the non-linear dynamics of flow and transport along the soil profile (Mohanty and Skaggs, 2001; Kim and Barros, 2002). Furthermore, DLNM offered enough flexibility to model a variety of dependencies in the time series dataset by selecting a suitable basis function. As an analogy, a DLNM is to a linear time series model (e.g. autoregressive model) just as a generalized linear model is to a linear model, as can be seen in Eq. 1.

In assessing lagged dependence, event scale patterns were of interest rather than large scale trends within the time series (Wilson et al., 2004). This required seasonal patterns to be addressed prior to applying the DLNM. This was done by fitting a loess function to the time series and then subtracting it from the original soil moisture values (Cleveland et al., 1990). Removal of seasonality was further justified by the scatterplot results (see Section 4.1). The influence of seasonality on the vertical soil moisture variability is indicated by clustering of observation points occurring within the same months (fig.4). De-seasonalized soil moisture values were used for identifying (de)coupled soil moisture conditions.

For consistency in modeling, the range of surface soil moisture values used was from 0-0.50 $cm^3cm^{-3}$. This was based on the highest surface soil moisture value encountered among the four sites. A lag value of up to 30 days was considered long enough to investigate delayed effects. This period also approximated the recurrence of heavy rainfall within the study sites. A spline function was the basis function chosen to represent the functional dependence and delayed effects as it offered flexibility to capture non-linearities. In addition, contributions from daily rainfall data were used to incorporate current and past meteorological conditions. This was applied as a covariate and was represented with an additional basis function. We only considered delayed effects in vertical flow as lateral movement is deemed negligible in a flat to slightly sloping terrain (Table.1). The analysis was performed in R software using *dlnm* (Gasparrini, 2011) and *mgcv* (Wood, 2006a) packages.

The following section concisely describes the mathematical formulation of a DLNM. However, the reader may choose to skip this section as the general description of the methods applied have already been given in the text above. For a more detailed explanation, readers are referred to Gasparrini et al. (2010) and Gasparrini et al. (2017).

To more formally describe a DLNM, let us first consider a general time series model, where outcomes $Y_t$ with $t = 1, \cdots, n$ can be described by:

$$g(\mu_t) = \alpha + \sum_{j=1}^{J} s_j(x_{tj}; \boldsymbol{\beta}_j) + \sum_{k=1}^{K} \gamma_k \boldsymbol{u}_t k \tag{1}$$

where $\mu \equiv E(Y)$, assumed to be derived from a Poisson distribution, and $g$ is a monotonic link function. The functions $s_j$ denote relationships between the variables $x_j$ and vector parameters $\boldsymbol{\beta}_j$. Other $u_k$ variables with predictors are included in coefficients $\gamma_k$ to specify their related effects. The relation between $x$ and $g(\mu)$ is represented by $s(x)$ through a basis function. The complexity of this estimated relationship depends on the type basis function chosen and its dimensions. In the presence of delayed effects, the outcome $Y$ at any time $t$ is explained by the past exposures $x_{t-l}$ with $l$ as the *lag* representing the elapsed time between exposure and response. The final goal of a DLNM is to simultaneously describe the dependency along both the predictor space and lag dimension. This is achieved by selecting two sets of basis functions that are combined to obtain the cross-basis functions (Gasparrini et al., 2010).

Within the DLNM framework, a response $Y_t$ at time $t = 1$ is based on lagged occurrences of predictor $x_t$, which is represented by vector $q_t = [x_{t-l_0}; \cdots; x_{t-L}]^T$. The minimum and maximum lags are given by $l_0$ and $L_T$, respectively. The function represents dependence through:

$$s(q, t) = s(x_{t, t-l_0}, \cdots, x_{t-L}) = \sum_{l=l_0}^{L} f \cdot w(x_{t-L}, l) \tag{2}$$

where $f \cdot w(x_{t-L}, l)$ represents the exposure-lag-response function, which is composed of two marginal functions: the exposure-response function $f(x)$ and lag-response function $w(l)$ in the space of the lag. Parameterization of $f$ and $w$ is achieved by application of the known basis functions to the vectors $q_t$ and $l$. The result can be expressed as matrices $\mathbf{R}$ and $\mathbf{C}$ with dimensions $(L - l_0 + 1) \times v_x$ and $(L - l_0 + 1) \times v_l$, respectively.

The cross basis function $s$ and parameterized coefficients $\boldsymbol{\eta}$ are given by:

$$s(x_{t, t-l_0}, \cdots, x_{t-L}; \boldsymbol{\eta}) = (1_{L-l_0+1}^T \boldsymbol{A}_t)\boldsymbol{\eta} = \boldsymbol{w}_t^T \boldsymbol{\eta} \tag{3}$$

The values of $\boldsymbol{w}$ are derived from $\boldsymbol{A_t}$, which is computed from the row-wise Kronecker product between matrices $\mathbf{R}$ and $\mathbf{C}$. The dependence is expressed through $\boldsymbol{w}$ and parameters $\boldsymbol{\eta}$. The cross-basis function represents the integral of $s(x, t)$ over the interval $[l_0, L]$, summing the contributions from the exposure history. The estimated dependence to specific exposure values is

determined by prediction of $\hat{\beta}$, called lag coefficients. The estimated $\hat{\beta}$ and covariance matrix $V(\hat{\beta})$ is given by:

$$\hat{\beta} = \boldsymbol{A}_x \hat{\boldsymbol{\eta}} \tag{4}$$

$$V(\hat{\beta}) = \boldsymbol{A}_x V(\hat{\boldsymbol{\eta}}) \boldsymbol{A}_x^T \tag{5}$$

A further extension to DLNM is the application of penalties for smoothness of the lag structure and shrinkage of lag coefficients to null at very high lags. These penalties were applied in the analysis using a second-order difference (Wood, 2006b) and varying ridge penalties (Obermeier et al., 2015; Gasparrini et al., 2017), respectively. Application of penalties was based on the assumption that, at higher lags, the lag coefficients become smaller and approach the null value.

### 3.3 Evaluating (de)coupled soil moisture values

Application of a DLNM resulted in the estimation of parameter $\hat{\beta}$ for each surface soil moisture value (Eq. 4 and 5). This indicated the strength of dependence between surface and subsurface soil moisture. Higher $\hat{\beta}$ values indicated stronger dependence or coupling between the two. Hence, we referred to $\hat{\beta}$ as the relative influence of surface soil moisture on subsurface values.

## 4 Results

### 4.1 Regression and Residuals analysis

The overall dependence between surface and subsurface given by the Spearman's rank coefficient ($R_s$) range from 0.746 to 0.866 (fig.4). However, even with a high overall dependence, variability is not uniform across the soil moisture range (fig.4). Except for SM13, increased variability is observed towards drier soil moisture values. Furthermore, the degree of variability also differs among the four sites. The most pronounced variability is observed at SM13 and the least at SM05. Clustering of observation points occurring within the same months indicate that seasonality dictates soil moisture values and impacts soil moisture variability. Rainfall events measured on the same day do not show a clear effect on surface and subsurface soil moisture dependence. Observations with higher rainfall intensities appear scattered in the plots (fig.4). In addition, the said observation points do not necessarily fall along the fitted functions or at the wet soil moisture region of the scatterplots. As lag is not considered, the impact of rainfall on variability is not fully captured in the scatterplots alone.

INSERT Fig.4 here

Assessment of the regression fit quality was performed by comparison using residual standard errors (RSE). The results for both linear and loess functions show highly similar values (fig.4). This indicates that, in this case, a linear function captures the relation between surface and subsurface values. Nevertheless, the more flexible loess function was preferred for further residuals analysis because of its slightly better model fit and, using only visual inspection of fig.4, it more closely approximates the calculated conditional mean.

Figure 5 shows the residual plots with lines of the cumulative residuals variance. The change in slope of the line is a feature consistent for all sites regardless of the magnitude of residual variance. The changes in variability are more clearly observed from the residuals than from the standard deviation bars in the scatterplots. The change in slope at $\theta_c$ is highlighted by the vertical dashed line. The Decoupled soil moisture range corresponds to the section of cumulative residuals variance line with a steeper slope. Specifically, the range of decoupled surface soil moisture values (in cm$^3$cm$^{-3}$) is 0.08-0.21 for SM05, 0.12-0.27 for SM09, 0.30-0.39 for SM13, and 0.08-0.12 for SM20. Except for SM13, the decoupled values are within the dry to intermediate soil moisture range. The cumulative residuals variance line for SM13 appears to increase exponentially with increasing surface soil moisture. This differs from the other three sites which show a distinct decrease in slope at increasing soil moisture values. For SM20, a second point is identified with a change in slope. The flat line starting from 0.24-0.28 cm$^3$cm$^{-3}$ indicates there is still lowered variance at the very wet soil moisture range.

The correlation between normalized variance and sample size yielded a value of -0.24 (fig.6). This low correlation magnitude confirms that the variance obtained for the soil surface moisture intervals was not strongly influenced by the sample size used.

INSERT Fig.6 here

INSERT Fig.7 here

## 4.2 Cross-correlation

Figure 7 shows cross-correlation values at the four sites. Maximum correlation occurs at -1 to -2 days lag, except at SM20. This translates to a 1-2 day lead of surface soil moisture values. For SM20, the maximum correlation occurs at positive lags. Correlation values from lag = 0 to lag = 10 are almost equal at SM20. Although this indicates leading subsurface values, it does not eliminate the possibility of having a lag between surface and subsurface values (see Section 5.2). Other factors may play a role in having leading subsurface values in the cross-correlation plots. Hence, SM20 was still analyzed for decoupling using DLNM.

## 4.3 Distributed lag model

Figure 8 shows the overall $\hat{\beta}$ for each surface soil moisture value with 5% and 95% confidence intervals in shaded gray regions. In order to identify a range that is decoupled, a threshold value ($\hat{\beta}_c$) must be specified. This value is comparable to the intermediate soil moisture $\theta_c$ identified from Section 4.1. The values of $\theta_c$ provided a suitable guide for identifying a threshold common to all four sites (Table 2). The corresponding $\hat{\beta}$ values obtained at $\theta_c$ were very close to 1, therefore, setting the threshold $\hat{\beta}_c = 1$ seemed a reasonable choice. This was preferred over the exact $\hat{\beta}$ at each $\theta_c$ since the latter was defined using exploratory methods at $lag = 0$. Using the chosen $\hat{\beta}_c = 1$, surface soil moisture values with $\hat{\beta} < 1$ are considered decoupled while those with $\hat{\beta} \geq 1$ are coupled.

Based on $\hat{\beta}_c$, the identified decoupled values are generally in the dry to intermediate soil moisture range (fig.8), except for SM13 where decouple values are at the wet range. Table 2 shows the decoupled values identified based on the selected $\hat{\beta}_c$. The behavior and trends of $\hat{\beta}$ also differ for each station. For instance, at SM05 and SM09, there is a general increase

in $\hat{\beta}$ from dry towards wet surface soil moisture values. SM20 also shows increasing $\hat{\beta}$ over a limited soil moisture range $(0.1 - 0.25\ cm^3cm^{-3})$. Outside this range, the estimated $\hat{\beta}$ values for SM20 were less than one and have very broad confidence intervals. Recall that the range used for DLNM was only for uniformity among the four study sites. The lack of or very few observations for very dry or very wet soil moisture conditions led to wider confidence intervals not only for SM20 but also for the other three sites. Compared to the three sites, the estimated $\hat{\beta}$ values for SM13 show decreasing values towards the wet soil moisture range ( $> 0.3\ cm^3cm^{-3}$ ). From the intermediate to dry soil moisture conditions, the values fluctuate around the designated $\hat{\beta}_c$.

INSERT Fig.8 here

INSERT Table 2 here

## 5 Discussion

### 5.1 Decoupled soil moisture values

Regression and residuals analyses show that there is an inherent vertical variability between surface and subsurface soil moisture values based on the lack of 1:1 correspondence between the two (fig.4). This inherent variability is also not uniform as higher variability is observed at certain soil moisture ranges. The cumulative residual variance plots (fig.5) clearly indicate the soil moisture values where vertical variability starts to become consistently larger. The increase in variability further translates to weak lagged dependence which we observe as low $\hat{\beta}$ values from DLNM. The increase in vertical variability and weakening of lagged dependence is what we considered as decoupling between the surface and subsurface soil moisture.

Both residuals analysis and DLNM were successful in identifying a decoupled soil moisture range and there is good agreement between the results from both. Three out of four sites show decoupled values in the dry to intermediate soil moisture range (fig.5 and Table 2). These results agree with the known range where decoupling is expected (Capehart and Carlson, 1997; Hirschi et al., 2014; Wilson et al., 2003). For SM05 and SM09, the intermediate soil moisture value, $\theta_c$ that marks when decoupling begins (Table 2) is close to that identified by Capehart and Carlson (1997). They obtained a value of 0.3 cm³cm⁻³ as the point below which decoupling begins. However, results for SM13 do not conform to the traditional concept of decoupling. This result is significant as it implies that decoupling may occur at any value and is not confined to dry soil moisture range.

The vegetation type at each site exerts some influence on the soil moisture variability and the resulting (de)coupled values. First, the vegetation type affects how much ground surface is directly exposed to atmospheric conditions. Forested areas and grass fields are almost fully covered by vegetation compared to a corn field where the crops are organized in equidistant rows. Vegetation or canopy cover will determine how atmospheric conditions affect the soil moisture values. For instance, the amount of intercepted precipitation and evaporation are both dependent on vegetation cover. This in turn will have direct impacts to the surface soil moisture dynamics at each of the sites. For comparison, the variability given by the standard deviation bars in fig.4 and variance in fig.5 at the cornfield (SM09) is higher compared to that of the grass field (SM05) or the forested area (SM20). In addition, the forested area (SM20) has the smallest range of soil moisture values among the four sites. This may be

due to the large intercepted rainfall by the forest canopy. Root water uptake (RWU) is another way by which vegetation affects soil moisture variability. RWU can have significant influence on the subsurface dynamics. The influence of RWU may vary for different vegetation types as it can be exerted over a range of depths, leading to differences in the resulting (de)coupled values.

Among the four sites, the subsurface trends observed for the 40 cm values at SM13 show consistently high values, which can be more pronounced during winter months. This resulted in decoupling during wet soil moisture conditions fig.8. This trend is different from the other three sites which only show a slight increase in the subsurface values. Further inspection of the time series data at SM13 reveals no sudden disturbance in the signal which could be attributed to errors in the sensor. Field investigation confirmed an increase in silt content at 40 cm compared to the upper layers. The increase in silt content promotes a decrease of hydraulic conductivity over depth that results in a slower vertical flow towards deeper layers. The presence of burrowing and hibernating animals was also observed at the site during winter. These create macropores which eventually alter the hydraulic properties of the soil (Kodešová et al., 2006; Beven and Germann, 2013). We infer that, at the measurement domain of the sensor, these burrows or macropores facilitated faster vertical flow to the subsurface. Alternatively, if the burrows produced voids around the measurement domain, this would result in lowered soil moisture or data gaps due to the loss of sensor to soil media contact. However, there were no gaps observed that coincided with the burrowing animals' period of hibernation. During precipitation events, soil moisture flowing from upper layers arrived more rapidly at 40 cm depths due to the presence of macropores. There it accumulated and flowed more slowly to deeper layers because of the low hydraulic conductivity promoted by the increase in silt content. The overall effect of these factors was the pronounced increase in soil moisture values at 40 cm compared to those at 5 cm during winter periods as observed from the time series dataset fig.2.

Site-specific characteristics at each station control the magnitude of variability as well as the range at which decoupling is observed. However, the occurrence of decoupling is independent of the magnitude of variability since it was observed from SM05 where variability is least up to SM13 where it is greatest. The methods applied in this study only identify conditions when decoupling occurs but do not explicitly determine its controls. Identification of controls for decoupling requires a separate analysis where mechanistic models or statistical approaches can be applied.

## 5.2 Assessing the use of lagged dependence for identifying decoupled conditions

To assess the applicability of the methods applied, we further discuss their strengths and weaknesses. We also present opportunities for further studies as well as foreseen limitations for other sites.

*Strengths:* The residuals analysis and DLNM methods allow quantification of a range of soil moisture values where decoupling occurs. This provides further extension to previous studies where decoupling is only described qualitatively. As seen from the results at the four sites, decoupling can occur at any soil moisture value, and is not confined to dry periods or ranges. Furthermore, by making no initial assumptions on data distributions and the type of functional relation and lag structure, the methods applied were considered robust. Non-linear functions were applied as they conform to the nonlinearity of water flow in the unsaturated zone. They can also handle a variety of bivariate dependence, even in cases where the relation is linear, as shown by the highly similar fit of the loess and linear functions in Section.4.1.

*Weaknesses:* The first aspect that needs to be further investigated is the selected $\hat{\beta}_c$ value for identifying the decoupled soil moisture range. Although the selection in this study was based on trends identified from time series datasets, the methods applied should be tested further using other datasets to confirm the suitability of $\hat{\beta}_c = 1$ for other depths and soil types. The choice of $\beta_c$ is crucial as it dictates which soil moisture values are expected to be decoupled. For instance, at the sites where decoupling occurs during dry conditions, a higher $\beta_c$ value would enlarge the decoupled range. A similar effect would be expected for the site with decoupling during wet conditions. However, a lower $\beta_c$ value could result to decoupling only during extreme soil moisture conditions (e.g very wet or very dry).

Another aspect to further examine is the use of cross-correlation for confirming the presence of leading surface soil moisture values. Results from SM20 show maximum correlation at positive lags which indicate leading subsurface values (fig.7). The weakness of using cross-correlation as a test for the presence of lag can be two-fold. First, cross-correlation can also capture the effect of subsurface dynamics such as groundwater influence and lateral flow. We infer that in SM20, subsurface dynamics dominates and masks the lag relation sought. An additional covariate representing subsurface dynamics was not included in the DLNM analysis since a dominant downward vertical flow was assumed. This assumption was based on the flat slopes encountered at SM20 (Table 1). Therefore, the occurrence of subsurface lateral flow or groundwater influence pose limitations to the applicability of DLNM for assessing decoupling. Second, cross-correlation is limited to evaluating linear lagged dependence and in incorporating non-linear lagged dependence can make the test more robust. Equivalent methods exist (e.g. mutual information content (Qiu et al., 2014)) but they are much more computationally demanding when the goal is simply to check for the existence of lag-lead relation.

*Opportunities:* In relation to utilizing remote sensing techniques, our results imply that the accuracy of estimating subsurface values from surface soil moisture can be greatly affected by vertical coupling. Lower variability and hence lower uncertainties are expected in the coupled soil moisture range. Assessment of decoupling can be used in combination with modeling studies as a preliminary method to determine the range where variability is expected to be higher. Furthermore, it can be helpful in assessing whether simulation results capture the variabilities observed in both the coupled and decoupled ranges. Taking decoupling into account can also assist in evaluating the necessity of complex models for simulating vertical soil moisture content.

For data assimilation (DA) applications, (de)coupling methods can be used for cross-comparison of the vertical coupling derived from DA model outputs with those observed from long term in situ measurements. This can aid in examining the adequacy of the assumed inherent connection between surface and subsurface values. As Kumar et al. (2009) pointed out, land surface models vary in their representation of the strength of this connection (e.g. weak or strong connection) which contributes the degree in which modeling results are improved. They also suggested that strong coupling is a more robust choice unless independent information suggests that a more decoupled surface-subsurface representation is more realistic. In this aspect, the analysis applied in this study could be a valuable tool in determining which type of surface-subsurface coupling is the more optimal choice. Furthermore, the assumed connection strength is adopted for the whole range of soil moisture values. The results of our analysis show that at any given site, decoupling will occur regardless of degree of soil moisture variability.

A variable coupling strength could be adopted based on the soil moisture range where decoupling is likely to occur as an alternative to the single value for the whole range.

INSERT Fig.9 here

Although the study focused on vertically discrete values, the results are also applicable for depth-average values commonly used in remote sensing and DA applications. This requires that the vertically discrete values adequately capture the overall dynamics within zone being investigated. In such a case, we infer that the translation to depth-averaged values would result in (de)coupled values that are close, but not identical, to the values obtained when only comparing two discrete depths. As an illustration, we calculated the depth-average values using all the available measurements at each site (i.e. 5, 10, 20 and 40 cm depth) following the formula from Qiu et al. (2014). Figure 9 (left) reveals highly similar dynamics for both discrete and depth-average values. Therefore, it can be expected that the results from a regression and DLNM analyses using depth-average values would be highly similar to the original results in fig.5 and fig.8. However, if the vertically discrete values insufficiently represent the subsurface dynamics, larger deviations in the resulting decoupled values can be expected.

*Limitations:* In this study, only meteorological factors were incorporated in the DLNM analysis since vertical movement was assumed to be the dominant flow mechanism. However, the subsurface can also be influenced by lateral movement or groundwater by capillary rise. In such scenarios, decoupling will not be limited to changes in surface conditions. For this, SM20 provides an excellent example. This station is located at the foot of a small hill (fig.2) where the occurrence of lateral subsurface movement is highly probable. This shows that although the analysis would be limited to smaller scales, or even a single point, recognition of regional setting is important for interpretation of results. In addition, subsurface dynamics can also be affected by capillary rise in areas with shallow groundwater. For future applications, the effect of both capillary rise and lateral movements to subsurface dynamics should be assessed and included in the DLNM analysis, but caution should be exercised when interpreting the results. Assessment of decoupling with DLNM is deemed more applicable to areas where the subsurface has insignificant groundwater influence and where vertical downward movement is the dominant flow mechanism.

## 6 Conclusion

The methods applied in this study allow for investigation of vertical soil moisture variability. More importantly, application of DLNM allowed for decoupled soil moisture range to be quantitatively identified. The results also reveal that decoupling is not confined to dry soil moisture range as implied by previous studies. The reasons for decoupling are manifold and controls for the dry soil moisture range may differ from those for the wet range. The results of this study have implications for remote sensing and data assimilation methods, especially for uncertainties related to the use of surface soil moisture to obtain integrated soil moisture values.

*Data availability.* The datasets for soil moisture were obtained from the Water Resource Department of ITC-Twente University. At the moment, the datasets are not publicly available. Access to the datasets may be granted upon request from the institute thru Prof. Rogier van der Velde, PhD (r.vandervelde@utwente.nl).

*Author contributions.* Coleen Carranza and Martine van der Ploeg initially conceptualized the idea for investigating the relation between surface and subsurface soil moisture values. Paul Torfs provided significant contributions to the statistical analysis applied. All three authors contributed to writing and editing of the manuscript.

*Competing interests.* No competing interest present

*Acknowledgements.* The authors are grateful for the Water Resources Department of ITC-Twente University, the Netherlands for sharing the datasets from their network. We thank the three anonymous reviews for providing critical insights that improved the manuscript.This work

is part of the research programme Optimizing Water Availability through Sentinel-1 Satellites (OWAS1S) with project number 13871 which is (partly) financed by the Netherlands Organisation for Scientific Research (NWO).

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

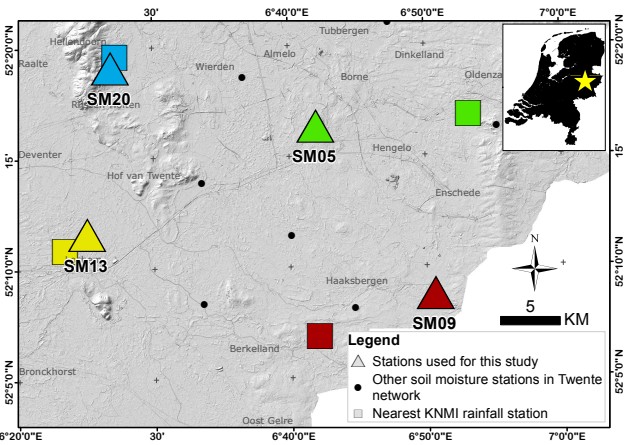

**Figure 1.** Location of study site in the eastern part of the Netherlands (inset). Triangles represent stations used within the Twente soil moisture and temperature monitoring network (Dente et al., 2011). Squares represent meteorological stations. Symbols with similar colors indicate the pair of measurements used for the analysis.

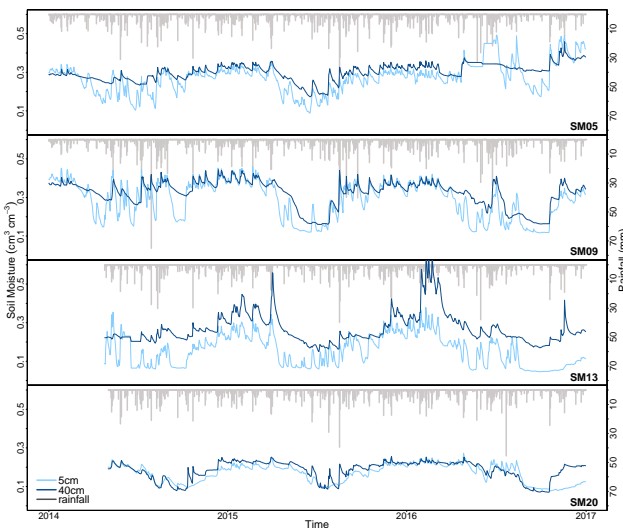

**Figure 2.** Time series plots of surface (5 cm in light blue) and subsurface (40 cm in dark blue) soil moisture. Vertical black bars at the top show daily precipitation data from the nearest KNMI station.

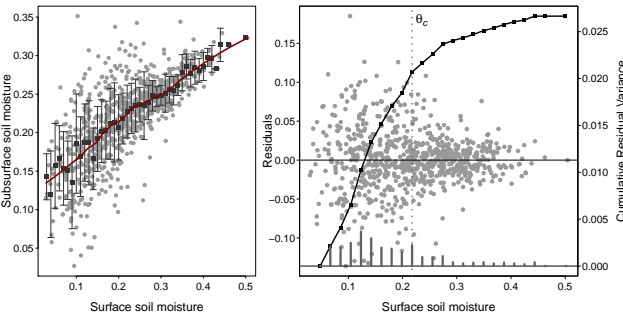

**Figure 3.** Schematic diagram using hypothetical soil moisture values to show vertical variability. Left: Scatterplot showing the trend with a fitted loess function. The variability can be seen using the standard deviation bars. Right: Scatterplot of the residuals from the fitted function. Soil moisture variability is visible from the variance given by the vertical bars and the cumulative residuals variance given by the black line. A change in variability at an intermediate soil moisture value is marked by a change in the slope of the cumulative variance line, indicated by $\theta_c$

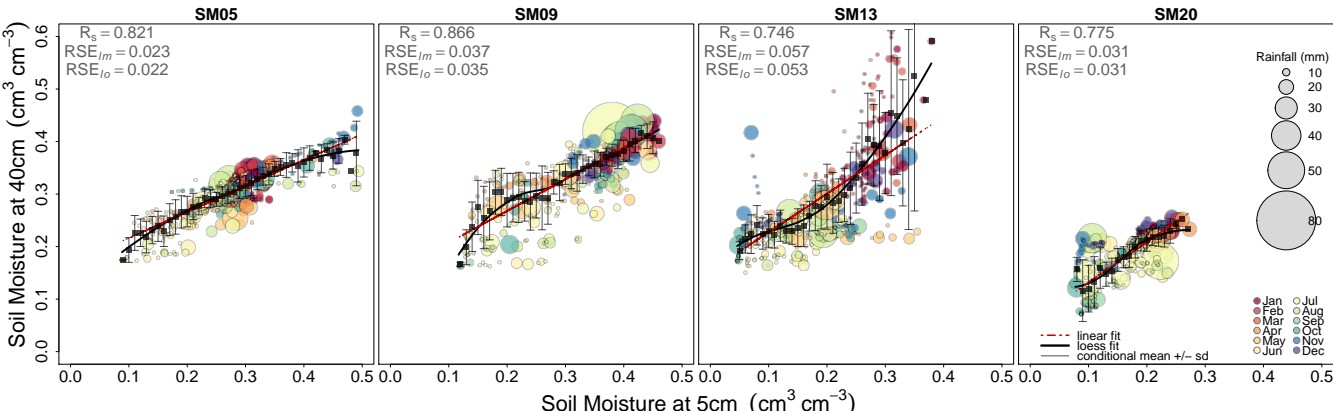

**Figure 4.** Scatter plots of 5 cm vs 40 cm soil moisture values at $lag = 0$. Colors correspond to the months in a year and sizes of points are proportional to rainfall intensity. Trends along the soil moisture range shown with the fitted loess function (black line). A linear function (red line) is also fitted for comparison. The overall dependence using Spearman's rank correlation $R_s$ is given in the upper left corner each plot. Residual standard errors (RSE) for loess (lo) and linear (lm) fits are also shown.

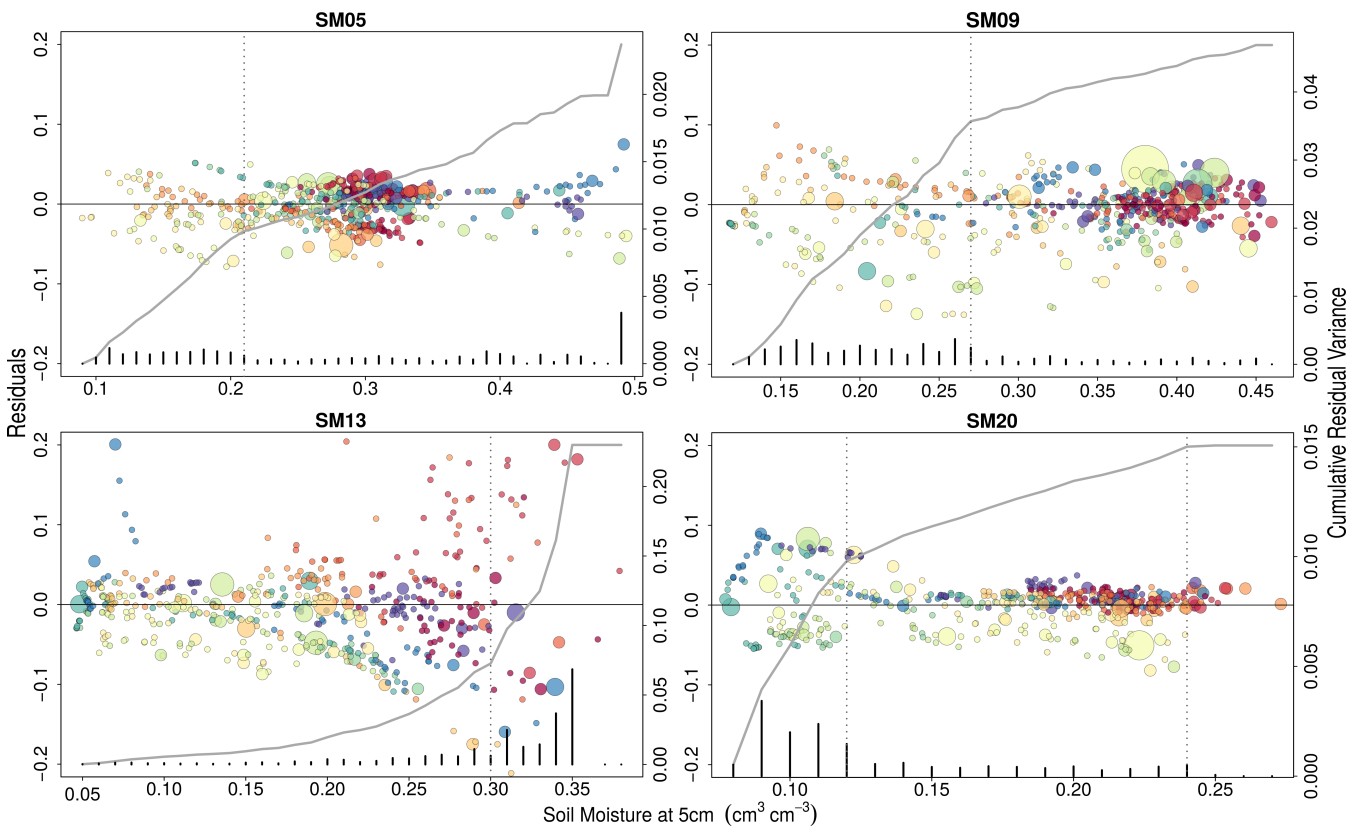

**Figure 5.** Residual variance plots from the fitted loess function. Vertical bars at the bottom of each plot represent the variance for every 0.1 cm³cm⁻³ interval. The cumulative residual variance (gray line) shows a change in slope at $\theta_c$, indicated by the vertical black dashed line. This separates the soil moisture values into a range with higher variance (steeper slope) and another range with lower variance (gentler slopes). The range with higher variance is considered decoupled, and vice versa.

**Table 1.** Summary of land cover descriptions at each station covering the period of 2014-2016. Soil descriptions and codes are based on BOFEK 2012 (Wosten et al., 2013). Both the slope and distance to nearest ditch were determined from 5m resolution DEM. Datasets are from 2016 and were obtained from the publicly available national topographic database of the Netherlands (TOPNL)

| Station No. | Land cover | BOFEK Soil description | Slope (degrees) | Aerial distance to nearest ditch (m) |
|---|---|---|---|---|
| SM05 | Grass | Loamy sandy soils with a thick cultivated layer(317) | 2.22 | 18.97 |
| SM09 | Corn | Weakly loamy sand soils with a thick cultivated layer(311) | 2.70 | 1.41 |
| SM13 | Grass | Weak silty soils (podsols)(304) | 1.0 | 17.09 |
| SM20 | Forest | Coarse sand (podsols)(320) | 2.30 | 875.26 |

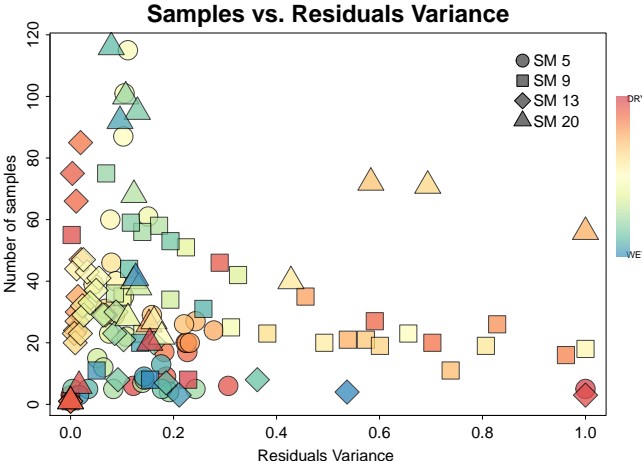

**Figure 6.** Scatterplot of sample size vs. normalized residual variance calculated for each 0.01 $cm^3 cm^{-3}$ interval. Colors indicate soil moisture conditions at each point. The plot of points indicate very weak linear dependence, which is further confirmed by a -0.24 correlation coefficient.

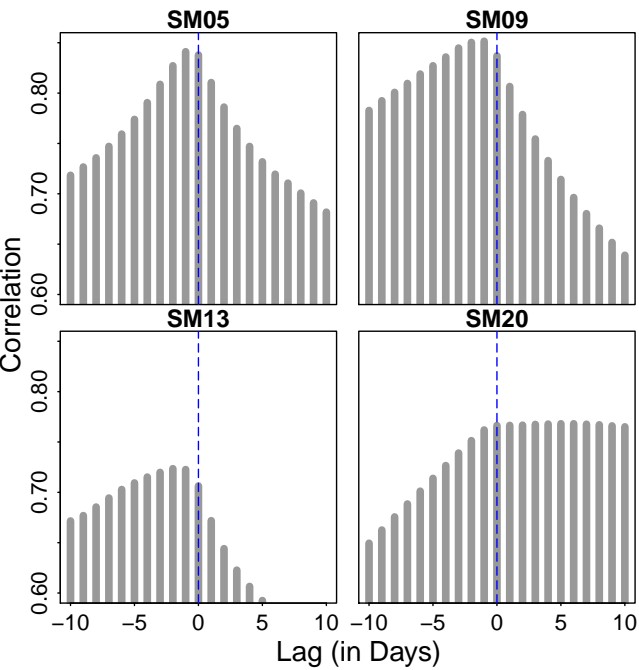

**Figure 7.** Cross-correlation plots of soil moisture values. The lagged surface soil moisture values at 5 cm are correlated with subsurface values at 40 cm. A 1-2 day lead of surface soil moisture is observed, except for SM20. This is indicated by having maximum the correlation values at lags of -1 to -2 days. At SM20, the maximum correlation occurs at positive lags.

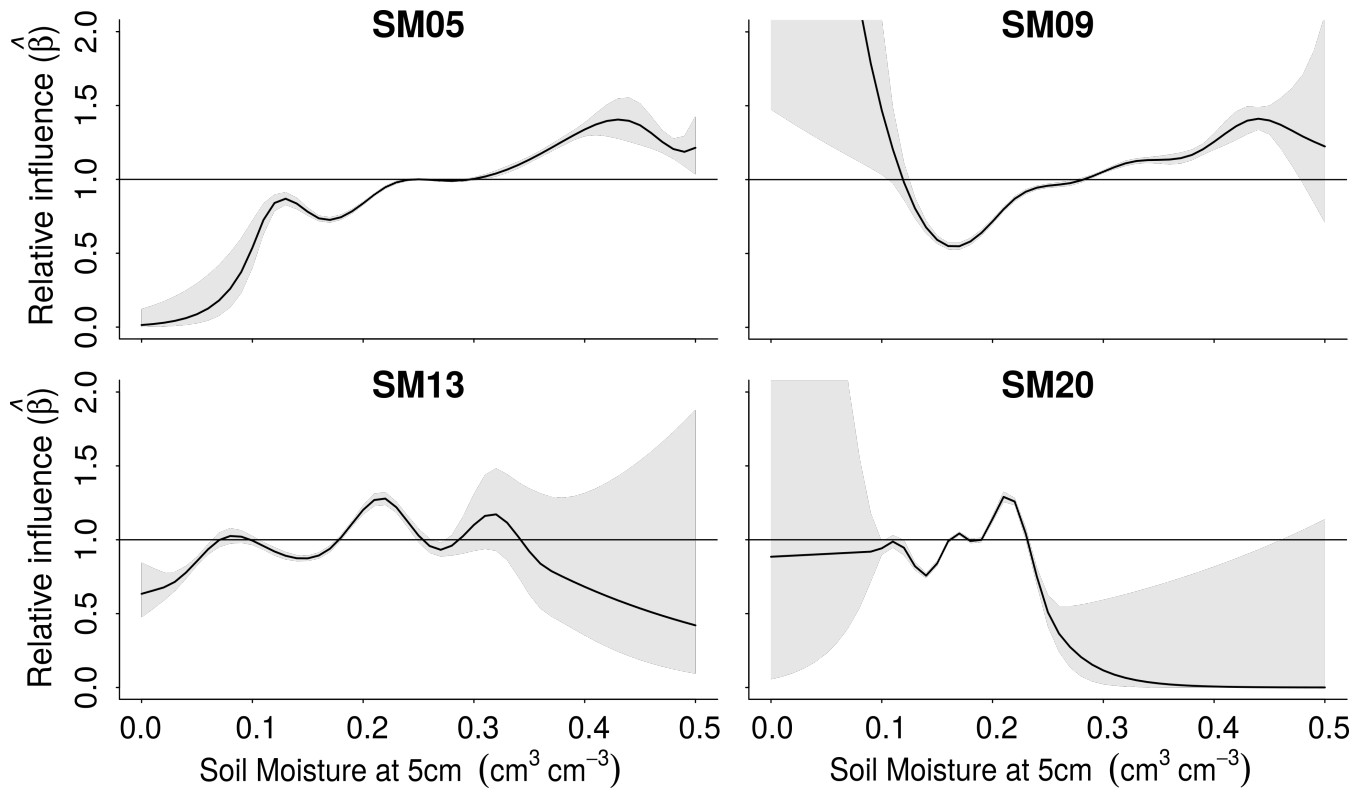

**Figure 8.** The relative influence of surface soil moisture on subsurface values obtained by summing the predicted $\hat{\beta}$ along the 30-day lag. The threshold value ($\hat{\beta}_c$) used to identify the decoupled range is indicated by the horizontal line. Surface soil moisture values below $\hat{\beta}_c$ are considered decoupled. The 5% and 95% confidence intervals of the predicted values are shown as shaded regions.

**Table 2.** List of surface soil moisture values (SSM in cm$^3$cm$^{-3}$) obtained from fig.5 and fig.8. SSM at $\theta_c$ in fig.5 were used to determine $\hat{\beta}$ in fig.8. A common threshold $\hat{\beta}_c$ was used for all sites since $\hat{\beta}$ are all close to 1. The resulting decoupled SSM values are shown in the fourth column.

| Station No. | SSM at $\theta_c$ | $\hat{\beta}$ at $\theta_c$ | Decoupled values using threshold $\hat{\beta}_c = 1$ |
|---|---|---|---|
| SM05 | 0.21 | 0.90 | <0.24 |
| SM09 | 0.27 | 0.97 | <0.28 |
| SM13 | 0.30 | 1.18 | >0.34 |
| SM20 | 0.12 | 0.94 | 0.16>SSM>0.23 |

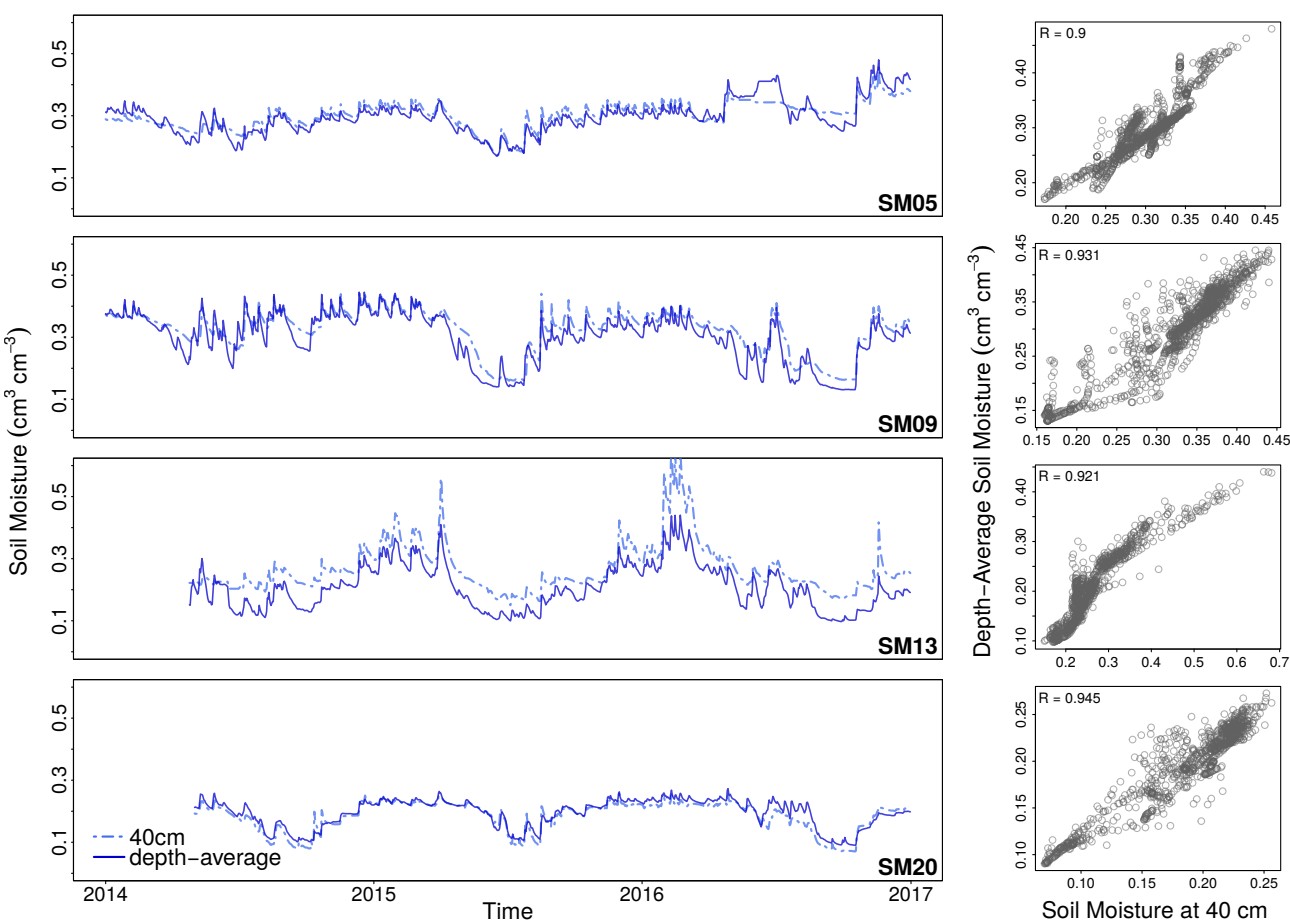

**Figure 9.** Subsurface soil moisture dynamics for vertically-discrete (40cm) and depth-average value. Left: Time series of soil moisture at 40 cm and depth-averaged values. The dynamics observed for depth-average values are highly similar to those at 40 cm. Right: Scatterplot showing that these two sets of values are highly correlated.