# Peer review of "Using lagged dependence to identify (de)coupled surface and subsurface soil moisture values"

_Hydrology and Earth System Sciences, 2017_

## Referee Comment (RC1) · Anonymous Referee #1 · 20 Dec 2017

The comment was uploaded in the form of a supplement:
https://www.hydrol-earth-syst-sci-discuss.net/hess-2017-651/hess-2017-651-RC1-supplement.pdf

---

## Referee Comment (RC2) · Anonymous Referee #2 · 22 Jan 2018

General comments: This study examines the coupling between near-surface and subsurface soil moisture at four sites in the Netherlands. Specifically the authors develop a methodology for determining when the two layers are decoupled, thereby providing an important analysis for surface soil moisture assimilation into models. The manuscript is very well written and the figures are well crafted. The use of a distributed lag nonlinear model for quantifying decoupled soil moisture ranges is novel and, as the authors point out, does not suffer from many of the assumptions and limitations of previously implemented methods. I recommend the manuscript for publication given appropriate consideration of the following concern.

Specific comments: The primary concern I have with the manuscript is the conclusion that the decoupled range is not limited to dry conditions. Evidence of this is provided at one of four sites (SM13), at which the authors confirm the presence of burrowing animals. Given the potential data quality issues at site SM13 and the fact that decoupling at the the other three sites was confined to the dry end of the soil moisture range, I believe the strong statements regarding soil moisture decoupling outside of dry conditions (i.e., section 5.1, line 9; section 5.2, lines 26-27) are not adequately supported by the results. Therefore I recommend the authors either soften this conclusion by adequately describing the uncertainty and lack of consistent supporting evidence, or assess why SM13 shows decoupling outside of dry conditions when the other three sites do not.

---

## Referee Comment (RC3) · Anonymous Referee #3 · 25 Jan 2018

This is a strong paper which makes a clear contribution. In particular, I appreciate the attempt to inject more rigor into the discussion surrounding vertically-coupling among multi-depth soil moisture measurements. I view the paper's contribution as being mainly methodological; however, some interesting preliminary conclusions regarding the occurrence of vertical de-coupling are presented (specifically, that – at least at one site – de-coupling is not limited to dry soil conditions).

The overall presentation of the manuscript is very good and the topic is of sufficient interest for HESS' readership. Therefore, I recommend publication following adequate response to the following minor points:

[Figure]

1) The manuscript would benefit from a more detailed description of exactly how their approach(es) can be used to improve the performance of a land data assimilation system. There are really two issues here. The first is a fundamental observability issue (i.e., what is the upper limit on how effectively surface soil moisture can be used to constrain sub-surface soil moisture given a perfect data assimilation issue). The second issue is the vertical accuracy of the assimilation paper. That is, even if you have high theoretical observability (i.e. high vertical coupling) at a particular point, you can still squander this potential constraint by using an assimilation model that does not properly represent this coupling. This is the model accuracy issue addressed previously by the Kumer et al. 2009 JHM paper (already cited in the original paper). To me, the second point is really the most important; however, addressing it requires the additional step of cross-comparing observed vertical coupling to the vertical coupling predicted by the assimilation model (when run off-line). It sounds like this is the ultimate intention of the authors; however, it is not explicitly spelled out in the current manuscript. So, in summary, I would encourage the authors to be more specific/detailed in describing exactly how these results can be used to improve the performance of a land data assimilation system.

2) The discussion in Section 3.2.2 is not very accessible. For example, equation (1) is introduced as describing the time series of "outcomes" $Y_t$, yet $Y_t$ only appears on the LHS of the equation as $g(E(Y))$. "g" is apparently a "monotonic link function" (??) and E is some kind of a an expectation operator (in space, in time, across an ensemble?). So it's hard for me to see how Yt is actually "described" here. In the next function sentence "s" is introduced as a "basis function" (?) and all this is before the actual DLNM model is introduced. I suspect that all this terminology is correct and adequate for an applied math audience; however, HESS readership will likely need a bit more help and conceptual background to get through this section. I'd strongly recommend that the authors revise/expand Section 3.2.2 with an eye towards making it more accessible to a general earth science audience. Especially the early part of the section between equations (1) and (2)...I struggled there to follow the authors'

approach.

3) The analysis here is based solely on vertically-discrete soil moisture measurements (i.e. soil moisture observed at a depth of 40 cm). However, in remote-sensing, modeling and data assimilation, soil moisture estimates reflect vertically-integrated values (within the measurement depth of the remote sensor or across the vertically-discrete soil layer specified in a land model). Will the transition between vertically-discrete versus vertically-averaged soil moisture values affect the applicability of these results in a modeling or data assimilation context? I would recommend more discussion on this point.

4) While the manuscript is generally well-written, it does contain a large number of minor English usage errors. Additional proof-reading is recommended.

---

## Author Comment (AC1) · 22 Feb 2018

Dear Reviewer,

We would like to express our gratitude for providing critical feedback to our manuscript. We attached a supplementary file with all the replies and the revisions applied to the text.

On behalf of all the authors, Coleen Carranza

Please also note the supplement to this comment:

[Figure]

https://www.hydrol-earth-syst-sci-discuss.net/hess-2017-651/hess-2017-651-AC1-supplement.pdf

---

## Author Comment (AC3) · 22 Feb 2018

Dear Reviewer,

We would like to express our gratitude for providing critical feedback to our manuscript. We attached a supplementary file with all the replies and the revisions applied to the text.

On behalf of all the authors,

Coleen Carranza

[Figure]

Please also note the supplement to this comment:
https://www.hydrol-earth-syst-sci-discuss.net/hess-2017-651/hess-2017-651-AC3-supplement.pdf

---

## Author Response (AR1)

**Point-by-point reply to review comments:**

We would like to thank the reviewers for providing critical feedback to our manuscript. We provided a reply to each comment which is followed by the revisions we made to the manuscript. The text for the reviewer comments are in grey while our replies and revisions are in black.

**Reviewer 1:**
Specific comments
Overall, the paper is well written but can be improved in what concerns the Methods section organization.

I have two main comments:
1) The first is related to the the choice of the value of the parameter $\beta c$. I think it is a key parameter for the study and should deserve further discussion (or results) on its impact on the results.
* * *
*We agree with the point raised by the reviewer on the parameter $\beta c$. We adapted the manuscript in Section 5.2 page 11 from line 1. The text now reads:*

*"Weaknesses: The first aspect that needs further investigation is the selected $\beta c$ value for identifying the decoupled soil moisture range. Although the selection in this study was based on trends identified from time series datasets, the methods applied should be tested further using other datasets to confirm the suitability of $\beta c = 1$ for other depths and soil types. The choice of $\beta c$ is crucial as it dictates which soil moisture values are expected to be decoupled. For instance, at the sites where decoupling occurs during dry conditions, a higher $\beta c$ value would enlarge the decoupled range. A similar effect would be expected for the site with decoupling during wet conditions. However, a lower $\beta c$ value could result to decoupling only during extreme soil moisture conditions (e.g very wet or very dry)."*

2) While the authors include different stations characterized by different vegetation cover (grass, corn and forest) its potential effect on the results is not examined in deep, especially for station SM20 which is located in the forested area, this effect could be significant. Can the authors provide further discussion on it?
* * *
*We agree with the reviewer that the discussion on the controls, such as the effect of vegetation was not very extensive. This is because our main focus for the study was to investigate the methodology applied to quantify coupled and decoupled soil moisture values. However, we do realize that providing further discussion on said topic would improve the quality of the discussion in the paper. We adapted the manuscript in 5.1 in page 9 starting from line 25*

*"The vegetation type at each site exerts some influence on the soil moisture variability and the resulting (de)coupled values. First, the vegetation type affects how much ground surface is directly exposed to atmospheric conditions. Forested areas and grass fields are almost fully covered by vegetation compared to a corn field where the crops are organized in equidistant rows. Vegetation or canopy cover will determine how atmospheric conditions affect the soil moisture values. For instance, the amount of intercepted precipitation and evaporation are both dependent on vegetation cover. This in turn will have direct impacts to the surface soil moisture dynamics at each of the sites. For comparison, the variability given by the standard deviation bars in fig.4 and variance in fig.5 at for the cornfield (SM09) is higher compared to that of the grass field (SM05) or the forested area (SM20). In addition, the forested area (SM20) has the smallest range of soil moisture values among the four sites. This may be due to the high amounts of intercepted rainfall by the forest canopy. Root water uptake (RWU) is another way by which vegetation affects soil moisture variability. RWU can have significant influence on the subsurface dynamics. The influence of RWU may vary for different vegetation types as it can be exerted over a range of depths, leading to differences in the resulting (de)coupled values."*

I have some additional and technical comments that I will list below in order of appearance in the manuscript:

1) Section 3.1: It is not clear to which figure the authors are referring to. Please point to a specific figure when describing graphic features or speak more in general. This section should contain method description and choices made for carrying out analysis.
* * *
*We adjusted the text in section 3.1 and added a schematic diagram which the text refers to.*

[Figure]

*Figure 3. Schematic diagram using hypothetical soil moisture values to show vertical variability. Left side of the figure shows the trend with a fitted loess function. The variability can be seen using the standard deviation bars. The right side shows the residuals of the fitted function. Soil moisture variability is more visible from the variance given by the vertical bars and the cumulative variance is given by the black line. A change in variability at an intermediate soil moisture value is marked by a change in the slope of the cumulative variance line, marked by θc*

2) Section 3.1: Define what vertical variability exactly means.
*We added a line in Section 3.1 in page 4 line 6:*
*"We referred to vertical variability as the unevenness or irregularity in soil moisture distribution within a certain depth of interest along the vertical profile, in this case up to depths of 40 cm"*

3) The cumulative variance is presented in Figure 4. Same as before, here the authors speak about slope but it is not clear at all for the reader to what slope they are referring to. Please define what cumulative residual variance means and clarify better all the related concepts.
    *-We adjusted the text in section 3.1 and added a schematic diagram, which is now figure 3, to refer to the parts of the graph being described in the text.  Please see response  for technical comment #1*

4) Section 3.2.2 Distributed lag model: It is not clear enough how the DLNM model is used with soil moisture observations. I suggest to describe the basic concepts here rather than mathematical details (which could be included in an appendix). Please try to simplify and clarify this section so as it can be more easily interpreted

*We realize the difficulty encountered by the reviewer in following the description of methods applied using the DLNM. We completely revised  3.2.2 section of the manuscript. The section now becomes:*

*"3.2.2 Distributed lag model*
*We incorporated delayed or lagged effects in evaluating the relation between surface and subsurface values, and eventually in determining (de)coupled values. It should be emphasized that the analysis was primarily focused on examining the trends and relations between surface and subsurface soil moisture. Moreover, it was not intended to contradict or replace other models for estimating soil moisture or examining its patterns.*

*A distributed lag non-linear model (DLNM) developed by Gasparrini et al. (2010) was applied to the 5 cm and 40 cm time series datasets at the study sites. Briefly, the model is capable of simultaneously representing both functional dependency and delayed response between exposure and response values. We considered surface soil moisture as the exposure values that produced delayed effects to the response values at the subsurface. A non-linear model was selected in order to capture the non- linear dynamics of flow and transport along the soil profile (Mohanty and Skaggs, 2001; Kim and Barros, 2002). Furthermore, DLNM offered enough flexibility to model a variety of dependencies in the time series dataset by selecting a suitable basis function. DLNM could be thought of as equivalent to a linear time series model (e.g. autoregressive model) just as a generalized linear model is equivalent to a linear model.*

*In assessing lagged dependence, event scale patterns were of interest rather than large scale trends within the time series (Wilson et al., 2004). This required seasonal patterns to be addressed prior to applying the DLNM. This was done by fitting a loess function to the time series and then subtracting it from the original soil moisture values (Cleveland et al., 1990). Removal of seasonality was further justified by the scatterplot results (see Section 4.1). The influence of seasonality on the vertical soil moisture variability is indicated by clustering of observation points occurring within the same months (fig.4). De-seasonalized soil moisture values were used for identifying (de)coupled soil moisture conditions.*

*For consistency in modeling, the range of surface soil moisture values used was from 0-0.50 cm 3 cm -3 . This was based on the highest surface soil moisture value encountered among the four sites. A lag value of up to 30 days was considered long enough to investigate delayed effects. This period also approximated the recurrence of heavy rainfall within the study sites. A spline function was the basis function chosen to represent the functional dependence as well as delayed effects as it offered*
*flexibility to capture non-linearities. In addition, contributions from daily rainfall data were used to incorporate current and past meteorological conditions. This was applied as a covariate and was represented with an additional basis function. We only considered delayed effects in vertical flow as lateral movement is deemed negligible in flat to slightly sloping terrain (Table.1). The analysis was performed in R software using dlnm (Gasparrini, 2011) and mgcv (Wood, 2006a) packages.*

*The following section concisely describes the mathematical formulation of a DLNM. However, the reader may choose to skip this section as the general methods applied have already been described in the text above. For a more detailed explanation, readers are referred to Gasparrini et al. (2010) and Gasparrini et al. (2017).*

==*####Some lines for the mathematical description of DLNM is not included here. The structure of this section was not revised#########*==

*3.3 Evaluating (de)coupled soil moisture values*
*Application of a DLNM resulted in the estimation of parameter β for each surface soil moisture value. This indicated the strength of dependence between surface and subsurface soil moisture. Higher β values indicated stronger dependence or coupling between the two. Hence, we referred to β as the relative influence of surface soil moisture on subsurface values."*

5) Pag. 6 lines 19-24: here the authors seem to anticipate the results of the paper about the value to be assigned to β c . However, I suggest to try to organize the paper in a way that the choice of β c is described in the results section (and supported by analysis).

*We agree with the reviewer that this paragraph is more suitable in the results section. This was moved to section 4.3  in page 7starting from line 24.*

6) Pag 8 lien 17-19. Clarify this sentence.
*We changed the text in page 8 line 17-19. It now reads as:*
*"For instance, at SM05 and SM09, there is a general increase in β from dry towards wet surface soil moisture values. SM20 also shows increasing β over a limited soil moisture range (0.1 −0.25 cm 3 cm -3 ). Outside this range, the estimated β values for SM20 were less than one and have very broad confidence intervals. Recall that the range used for DLNM was only for uniformity among the four study sites. The lack of or very few observations for very dry or very wet soil moisture conditions led to wider confidence intervals not only for SM20 but also for the other three sites. Compared to the three sites, the estimated β values for SM13 show decreasing values towards the wet soil moisture range ( > 0.3 cm 3 cm -3 ). From the intermediate to dry soil moisture conditions, the values fluctuate around the designated β̂c ."*

7) Pag 8 line 11. Replace 40cm with "40 cm"

*This was now changed. The whole manuscript was also reviewed for similar errors.*

8) Pag 9 . Line 25. T is missing.
*A "T"' is added in the beginning of the sentence. There was another round of proof reading to check for similar textual and grammatical errors.*

**Reviewer 2:**
General comments: This study examines the coupling between near-surface and sub-surface soil moisture at four sites in the Netherlands. Specifically the authors develop a methodology for determining when the two layers are decoupled, thereby providing an important analysis for surface soil moisture assimilation into models. The manuscript is very well written and the figures are well crafted. The use of a distributed lag non- linear model for quantifying decoupled soil moisture ranges is novel and, as the authors point out, does not suffer from many of the assumptions and limitations of previously implemented methods. I recommend the manuscript for publication given appropriate consideration of the following concern.

Specific comments: The primary concern I have with the manuscript is the conclusion that the decoupled range is not limited to dry conditions. Evidence of this is provided at one of four sites (SM13), at which the authors confirm the presence of burrowing animals. Given the potential data quality issues at site SM13 and the fact that decoupling at the the other three sites was confined to the dry end of the soil moisture range, I believe the strong statements regarding soil moisture decoupling outside of dry conditions (i.e., section 5.1, line 9; section 5.2, lines 26-27) are not adequately supported by the results. Therefore I recommend the authors either soften this conclusion by adequately describing the uncertainty and lack of consistent supporting evidence, or assess why SM13 shows decoupling outside of dry conditions when the other three sites do not.
* * *
*We acknowledge the reviewer's concern about the occurrence of decoupling at wet soil moisture conditions. Indeed, compared to the other three sites, this result is intriguing. We decided to follow the reviewer's recommendation to asses why SM13 shows decoupling outside conditions. In our view, the occurrence of decoupling during wet conditions is plausible as certain combinations of atmospheric, biological, and pedological conditions can promote this. We elaborated further on whether burrowing animals may have compromised the quality of the dataset at SM13. We argue that the creation of macropores by burrows did not result in lowered data quality as this would result in subsurface soil moisture dynamics that is opposite of what is observed from the time series datasets. We expanded the discussion in Section 5.1 in page 10 starting from line 4.*

*"Among the four sites, the subsurface trends observed for the 40 cm values at SM13 show consistently high values, which can be more pronounced during winter months. This resulted in decoupling during wet soil moisture conditions fig.8. This trend is different from the other three sites which only show a slight increase in the subsurface values. Further inspection of the time series data at SM13 reveals no sudden disturbance in the signal which could be attributed to errors in the sensor. Field investigation confirmed an increase in silt content at 40 cm compared to the upper layers. The increase in silt content promotes a decrease of hydraulic conductivity over depth that results in a slower vertical flow towards deeper layers. The presence of burrowing and hibernating animals was also observed at the site during winter. These create macropores which eventually alter the hydraulic properties of the soil (Kodešová et al., 2006; Beven and Germann, 2013). We infer that, at the measurement domain of the sensor, these burrows or macropores facilitated faster vertical flow to the subsurface. Alternatively, if the burrows produced voids around the measurement domain, this would result in lowered soil moisture or data gaps due to the loss of sensor to soil media contact. However, there were no gaps observed that coincided with the burrowing animals' period of hibernation. During precipitation events, soil moisture flowing from upper layers arrived more rapidly at 40 cm depths due to the presence of macropores. There it accumulated and flowed more slowly to deeper layers because of the low hydraulic conductivity promoted by the increase in silt content. The overall effect of these factors was the pronounced increase in*

*soil moisture values at 40 cm compared to those at 5 cm during winter periods as observed from the time series dataset fig.2."*

**Reviewer 3:**

This is a strong paper which makes a clear contribution. In particular, I appreciate the attempt to inject more rigor into the discussion surrounding vertically-coupling among multi-depth soil moisture measurements. I view the paper's contribution as being mainly methodological; however, some interesting preliminary conclusions regarding the occurrence of vertical de-coupling are presented (specifically, that – at least at one site – de-coupling is not limited to dry soil conditions). The overall presentation of the manuscript is very good and the topic is of sufficient interest for HESS' readership. Therefore, I recommend publication following adequate response to the following minor points:

1) The manuscript would benefit from a more detailed description of exactly how their approach(es) can be used to improve the performance of a land data assimilation system. There are really two issues here. The first is a fundamental observability issue (i.e., what is the upper limit on how effectively surface soil moisture can be used to constrain sub-surface soil moisture given a perfect data assimilation issue). The second issue is the vertical accuracy of the assimilation paper. That is, even if you have high theoretical observability (i.e. high vertical coupling) at a particular point, you can still squander this potential constraint by using an assimilation model that does not properly represent this coupling. This is the model accuracy issue addressed previously by the Kumar et al. 2009 JHM paper (already cited in the original paper). To me, the second point is really the most important; however, addressing it requires the additional step of cross-comparing observed vertical coupling to the vertical coupling predicted by the assimilation model (when run off-line). It sounds like this is the ultimate intention of the authors; however, it is not explicitly spelled out in the current manuscript. So, in summary, I would encourage the authors to be more specific/detailed in describing exactly how these results can be used to improve the performance of a land data assimilation system.
* * *
*We thank the reviewer for pointing out this aspect. Our intention with this paper was to develop a data driven assumption free method. We agree completely with the cross-comparison of observed vs modeled vertical coupling, but as the focus in this paper is indeed on the methodology itself we discussed to what extent such a comparison should be included in this paper. Nonetheless, based on the suggestion of the reviewer we revised the manuscript largely in Section 5.2 to accommodate the reviewers suggestion in page 11 starting from line 26*

*"For data assimilation (DA) applications, the applied (de)coupling methods can be used for cross-comparison of the vertical coupling derived from DA model outputs with those observed from long term in situ measurements. This can aid in examining the adequacy of the assumed inherent connection between surface and subsurface values. As Kumar et al. (2009) pointed out, land surface models vary in their representation of the strength of this connection (e.g. weak or strong connection) which contributes the degree in which modeling results are improved. They also suggested that strong coupling is a more robust choice unless independent information suggests that a more decoupled surface-subsurface representation is more realistic. In this aspect, the analysis applied in this study could be a valuable tool in determining which type of surface-subsurface coupling is the more optimal choice. Furthermore, the assumed connection strength is adopted for the whole range of soil moisture values. The results of our analysis show that at any given site, decoupling will occur regardless of degree of soil moisture variability. We suggest that perhaps a variable coupling strength could be adopted based on the soil moisture range where decoupling is likely to occur rather than a single value for the whole range."*

2) The discussion in Section 3.2.2 is not very accessible. For example, equation (1) is introduced as describing the time series of "outcomes" $Y\_t$, yet $Y\_t$ only appears on the LHS of the equation as $g(E(Y))$. "g" is apparently a "monotonic link function" (??) and E is some kind of a an expectation operator (in space, in time, across an ensemble?). So it's hard for me to see how Yt is actually "described" here. In the next function sentence "s" is introduced as a "basis function" (?) and all this is

before the actual DLNM model is introduced. I suspect that all this terminology is correct and adequate for an applied math audience; however, HESS readership will likely need a bit more help and conceptual background to get through this section. I'd strongly recommend that the authors revise/expand Section 3.2.2 with an eye towards making it more accessible to a general earth science audience. Especially the early part of the section between equations (1) and (2)...I struggled there to follow the authors' approach.
* * *
*Similar response to reviewer #1, comment #4.*
*We realize the difficulty encountered by the reviewer in following the description of methods applied using the DLNM. We completely revised  3.2.2 section of the manuscript.*

[revised manuscript text omitted]

*====Some lines for the mathematical description of DLNM is not included here. The structure of this section was not revised=========*

*3.3 Evaluating (de)coupled soil moisture values*
*Application of a DLNM resulted in the estimation of parameter β for each surface soil moisture value. This indicated the strength of dependence between surface and subsurface soil moisture. Higher β values indicated stronger dependence or coupling between the two. Hence, we referred to β as the relative influence of surface soil moisture on subsurface values."*

3) The analysis here is based solely on vertically-discrete soil moisture measurements (i.e. soil moisture observed at a depth of 40 cm). However, in remote-sensing, modeling and data assimilation, soil moisture estimates reflect vertically-integrated values (within the measurement depth of the remote sensor or across the vertically-discrete soil layer specified in a land model). Will the transition between vertically-discrete versus vertically-averaged soil moisture values affect the applicability of these results in a modeling or data assimilation context? I would recommend more discussion on this point.
* * *
*We thank the reviewer for raising this point. We added another paragraph in section 5.2  in page 12 starting from line 4 to tackle this topic.*

*"Although the study focused on vertically discrete values, the results are also applicable for depth-average values commonly used in remote sensing and DA applications. This requires that the vertically discrete values adequately capture the overall dynamics within zone being investigated. In such a case, we infer that the translation to depth-averaged values would result in (de)coupled values that are close, but not identical, to the values obtained when only comparing two discrete depths. As an illustration, we calculated the depth-average values using all the available measurements at each site (i.e. 5, 10, 20 and 40 cm depth) following the formula from Qiu et al. (2014). Figure 9 (left) reveals highly similar dynamics for both discrete and depth-average values. Therefore, it can be expected that the results from a regression and DLNM analyses using depth-average values would be highly similar to the original results in fig.5 and fig.8. However, if the vertically discrete values insufficiently represent the subsurface dynamics, larger deviations in the resulting decoupled values can be expected."*

[Figure]

Figure 9. *Subsurface soil moisture dynamics from vertically-discrete (40cm) and depth-average value. Left: Time series of soil moisture at 40 cm and depth-averaged values. The dynamics observed for depth-average values are highly similar to those at 40 cm. The scatterplots on the right further confirms that these two sets of values are highly correlated.*

4) While the manuscript is generally well-written, it does contain a large number of minor English usage errors. Additional proof-reading is recommended.
*There was another round of proof reading to check for textual and grammatical errors.*

**List of changes made to the manuscript:**

1. Added  sentences or paragraphs to methods, results, and discussion sections based the comments from the reviewers.

2. Added figures to methods and discussion based on the reviewer comments

3. Changed some text  after re-checking the grammar and spelling of the  whole manuscript

[revised manuscript text omitted]